# Lipid-Lowering and Antioxidant Effects of Self-Assembled Astaxanthin–Anthocyanin Nanoparticles on High-Fat *Caenorhabditis elegans*

**DOI:** 10.3390/foods13040514

**Published:** 2024-02-07

**Authors:** Deyang Yu, Meng Guo, Mingqian Tan, Wentao Su

**Affiliations:** 1State Key Laboratory of Marine Food Processing and Safety Control, Dalian Polytechnic University, Dalian 116034, China; 2Academy of Food Interdisciplinary Science, Dalian Polytechnic University, Dalian 116034, China; 3National Engineering Research Center of Seafood, Dalian Polytechnic University, Dalian 116034, China; 4Collaborative Innovation Center of Seafood Deep Processing, Dalian Polytechnic University, Dalian 116034, China

**Keywords:** astaxanthin, anthocyanin, *Caenorhabditis elegans*, high-fat diet, antioxidant, lipid-lowering, microfluidic chip, Mannich reaction

## Abstract

Obesity has become a serious global public health risk threatening millions of people. In this study, the astaxanthin–anthocyanin nanoparticles (AXT-ACN NPs) were used to investigate their effects on the lipid accumulation and antioxidative capacity of the high-sugar-diet-induced high-fat *Caenorhabditis elegans* (*C. elegans*). It can be found that the lifespan, motility, and reproductive capacity of the high-fat *C. elegans* were significantly decreased compared to the normal nematodes in the control group. However, treatment of high-fat *C. elegans* with AXT-ACN NPs resulted in a prolonged lifespan of 35 days, improved motility, and a 22.06% increase in total spawn production of the nematodes. Furthermore, AXT-ACN NPs were found to effectively extend the lifespan of high-fat *C. elegans* under heat and oxidative stress conditions. Oil-red O staining results also demonstrated that AXT-ACN NPs have a remarkable effect on reducing the fat accumulation in nematodes, compared with pure astaxanthin and anthocyanin nanoparticles. Additionally, AXT-ACN NPs can significantly decrease the accumulation of lipofuscin and the level of reactive oxygen species (ROS). The activities of antioxidant-related enzymes in nematodes were further measured, which revealed that the AXT-ACN NPs could increase the activities of catalase (CAT), superoxidase dismutase (SOD), and glutathione peroxidase (GSH-Px), and decrease the malondialdehyde (MDA) content. The astaxanthin and anthocyanin in AXT-ACN NPs showed sound synergistic antioxidation and lipid-lowering effects, making them potential components in functional foods.

## 1. Introduction

Obesity is a worldwide and evolving health issue that is associated with excessive fat accumulation and has been proven to lead to type II diabetes mellitus, cardiovascular diseases, reproductive failure, certain types of cancer, and premature death [1,2,3]. The World Health Organization (WHO) predicts that about 167 million people may be less healthy due to obesity and overweight [4]. To date, many interventions involving pharmacotherapy and dieting have been applied to prevent and control obesity [5,6,7]. However, the side effects of these strategies may cause more potential health risks which hinder their further application [8]. Subsequently, the development of facile and practical approaches for anti-obesity is urgently desirable. In this context, tremendous efforts have been devoted to employing natural bioactive compounds to develop functional food supplements that may complement traditional strategies for obesity management [2,9].

Oxidative stress is a harmful factor that plays a significant role in the development of obesity-related disorders [10]. Obese people exhibit higher levels of oxidative stress and poorer antioxidant defenses compared to lean people [11]. Astaxanthin (3,3′-dihydroxy-β-carotene-4,4′-dione), a marine carotenoid with high natural abundance, has been reported to be a high-quality functional food component with strong antioxidant activity and lipid-lowering capacity due to its long-chain conjugated double bond [12,13,14]. Wang et al. proposed that the AXT derived from *Haematococcus pluvialis* can alleviate obesity by reducing the content of total free fatty acids (FFAs), triacylglycerols, and cholesterol esters and modulating the gut microbiota of mice fed with a high-fat diet [12]. Choi et al. investigated the effects of astaxanthin on lipid profiles and oxidative stress in overweight and obese Korean adults, and they discovered that supplementary AXT has positive effects by reducing low-density lipoprotein cholesterol, apolipoprotein B, and oxidative stress biomarkers, including MDA, SOD, isoprostane, and total antioxidant capacity [15]. However, the low water solubility, stability and bioaccessibility of dietary AXT have restrained its further application in functional foods [16]. Therefore, it is highly desirable to apply advanced nanocarriers for improved bioavailability of AXT. In our previous work, astaxanthin–anthocyanin nanoparticles (AXT-ACN NPs) were successfully prepared by a microfluidic strategy through the Mannich reaction, which has been proved to have enhanced stability under high temperature, low pH, and UV irradiation [17]. Given anthocyanin exhibits great capability in lipid lowering and good antioxidant activity, it can be speculated that AXT-ACN NPs would have a positive effect on alleviating the health risks caused by obesity [18].

Herein, high-fat *C. elegans* were cultured by feeding a high-sugar diet and further used as a model organism to evaluate the impact of AXT-ACN NPs on the physiological indicators of the nematodes. After treatment with AXT-ACN NPs, the lifespan, motility, and total spawn production of high-fat nematodes were significantly increased. Furthermore, the AXT-ACN NPs were proven to have excellent capability in reducing the fat content, lipofuscin content, and ROS level of high-fat nematodes. In the end, the content of triglyceride (TG), non-esterified fatty acids (NEFA), and antioxidant-related enzymes in nematodes were carefully studied, which validated that AXT-ACN NPs can reduce the content of free fat acids in high-fat nematodes, reduce fat accumulation, and show great antioxidant ability, making them facile food-grade supplements and potential alternatives to weight loss medications.

## 2. Materials and Methods

### 2.1. Materials

Anthocyanin was purchased from Tianjin Jianfeng Natural products R & D Co., Ltd. (Tianjin, China); lysine was purchased from Shanghai aladdin Biochemical Technology Co., Ltd. (Shanghai, China); calcium chloride, agar, cholesterol, and peptone were purchased from Shanghai Macklin Biochemical Technology Co., Ltd. (Shanghai, China); NaOH and NaCl were purchased from Sinopharm Chemical Reagent Co., Ltd. (Shanghai, China); an Oil Red O staining kit was purchased from Beijing Leagene Biotechnology Co., Ltd. (Beijing, China); and TG assay kits, NEFA assay kits, MDA, and CAT assay kits were purchased from Nanjing Jiancheng Bioengineering Institute (Nanjing, China). All chemicals were used without purification.

### 2.2. Preparation of AXT-ACN NPs

The synthesis of AXT-ACN NPs was based on our previous report [17]. Briefly, 1 mg/mL of AXT ethanolic solution was used as the internal phase; 0.75 mg/mL of lysine aqueous solution and 60 μL formaldehyde served as the mesophase; and 3 mg/mL ACN aqueous solution was used as the external phase. Subsequently, the internal, intermediate, and external phase were transferred into 1 mL syringes. The microfluidic chip was first pumped with the external phase with a flow rate of 10 mL h^−1^, and then both the intermediate and external phase were pumped at the rate of 5 mL h^−1^. The output channel of the microfluidic chip was applied to collect the mixed solution. The ethanol was removed by rotary evaporator at 45 °C, and the AXT-ACN NPs were obtained via lyophilization. The ACN nanoparticles (ACN NPs) were prepared through the same approach without AXT.

### 2.3. Caenorhabditis Elegans Culture and Synchronization

The nematode growth medium (NGM) was prepared before culturing. Briefly, 17 g of agar, 3 g of NaCl, and 2.5 g of tryptone were dissolved into a mixed solution of 2.5 mL of 1 M phosphate buffer and 97.5 mL of distilled water. Subsequently, the mixture was sterilized in a 121 °C autoclave, which was cooled to 60 °C in a water bath for further use. Then, 1 mL of 1 M CaCl_2_ solution (sterile), 1 mL of 1 M MgSO_4_ solution (sterile), and 1 mL of sterile cholesterol (5 mg/mL in ethanol) solution were transferred to the solution prepared in the last step. The well-mixed solution was then poured into the Petri dish, and the NGM was successfully prepared.

*C. elegans* type N2 nematodes were fed in NGM with *Escherichia coil* (*E. coli*) OP50 as the food source at 20 °C. Synchronized *C. elegans* were obtained via the following steps. Firstly, gestational *C. elegans* were washed with M9 buffer (5 g L^−1^ NaCl, 3 g L^−1^ KH_2_PO_4_, 15.12 g L^−1^ Na_2_HPO_4_·12 H_2_O and 0.25 g L^−1^ MgSO_4_) and collected into a 10 mL sterile centrifuge tube. The supernatant was discarded after standing for 2 min, and the *C. elegans* was further rinsed several times with M9 solutions. Subsequently, 200 μL bleach solution (4 mL of NaClO solution and 5 mL of 1 mol L^−1^ NaOH solution) was added and centrifuged at 4000 rpm for 1 min. After that, M9 buffer containing nematodes was dropped onto a new NGM agar plate (containing *E. coli* OP50) at 20 °C to obtain L4-larval nematodes.

In the control group, the nematodes were cultured from eggs on NGM plates containing OP50 only. In the high-fat (HF) group, the nematodes were induced on NGM plates containing *E. coli* OP50 and 10 mM glucose for 5 days. Then, 1 mg mL^−1^ AXT, ACN NPs, and AXT-ACN NPs were mixed with *E. coli* OP50, respectively, to feed high-fat nematodes for 48 h, which were indexed as the experimental groups.

### 2.4. Lifespan Assay

Lifespan analysis was performed by the same assay for all groups (the control group, HF group, AXT group, ACN NPs group, and AXT-ACN NPs group). The dead and surviving age-synchronized young L4 larvae nematodes were counted and recorded every five days until the nematodes died completely. Nematodes that failed to respond to a gentle stimulus using a metal wire were considered dead.

### 2.5. Reproduction Assay

The reproduction assay was performed according to the method reported by Wang et al. with modification [19]. Briefly, the synchronized nematodes (10 worms per plate) in each group were shifted to fresh NGM plates every other day. After the worms were transferred to new plate, the old plate was still cultured in 20 °C for another 24 h, and the total number of offspring within 7 days was counted.

### 2.6. Head Swing Assay

The head swing assay of nematodes were carried out based on Chen’s method with slight modification [20]. Briefly, after being incubated for 5 days, the synchronized nematodes in each group were selected and transferred into a new NGM medium, respectively, which was further added with M9 buffer. After one minute of recovery, the number of head swings within one minute was recorded under a microscope. The number of head swings of the nematodes in each group was measured on days 0, 2, 4, 6, 8 and 10. A head swing was defined as the nematode body bending length being greater than or equal to half of the nematode body length.

### 2.7. Stress Resistance Assay

A stress resistance assay was conducted based on the method reported by Sun et al. and Chen et al. with minor modifications [20,21]. The thermotolerance assay was performed on the 4-day-old worms in different treatments after synchronization, and the nematodes were transferred to 37 °C incubator. For the oxidative stress assay, the 4-day-old worms in different treatments after synchronization were shifted to a fresh NGM plate with 10 mmol L^−1^ H_2_O_2_ to induce oxidative stress. In both assays, the number of survived nematodes were counted and recorded every two hours until all the nematodes were dead.

### 2.8. Oil Red O Staining of C. elegans

The lipid accumulation in *C. elegans* was quantified by Oil Red O staining. Additionally, the procedure was slightly modified according to Xie’s report [5]. At first, the synchronized nematodes in each group were collected and rinsed three times with M9 buffer, followed by being fixed in 4% paraformaldehyde for 20 min. Subsequently, the mixture was centrifuged and the supernatant was discarded. After being cultured with Oil Red O for 2 h, the nematodes were washed with M9 buffer three times to remove the dye. Finally, the stained worms were observed by an inverted microscope equipped with a digital camera.

### 2.9. Measurement of Lipofuscin Accumulation

This assay was performed according to the previously described method with minor modifications [22]. The synchronized nematodes treated with different conditions in each group were transferred into new plates daily in the first five days. From the 6th day to the 10th day, the nematodes were shifted to new plates every two days, respectively. On the 10th day of adulthood, the nematodes from each group were transferred on the 2% agarose pads on glass slides coated with 10 μL levamisole hydrochloride in M9 buffer for anesthetization. The images of lipofuscin autofluorescence were obtained under an inverted fluorescence microscope. The relative fluorescence intensity of the nematodes was quantified using the Image J software (1.54f).

### 2.10. Determination of the Reactive Oxygen Species (ROS) Level

Accumulation of ROS in *C. elegans* was measured by using 2,7-dichlorodihydrofluorescein diacetate (DCFH-DA) based on the method reported by Li et al. with slight modification [23]. Briefly, the L4-phase synchronous nematodes in each group were selected and washed with M9 buffer three times before incubating with DCFH-DA solution at 37 °C for 30 min. After that, the nematodes were washed with M9 buffer three times and visualized under a fluorescence microscope (485 nm excitation and 530 nm emission). The fluorescence intensity of the nematodes was analyzed using Image J software (1.54f).

### 2.11. Measurement of TG, NEFA, GSH-PX, SOD, CAT, and MDA Levels

In each group, the synchronized L4 larvae nematodes were collected and washed with M9 buffer three times and further treated with ultrasonication. The protein content was quantified using a BCA Protein Quantification Kit (Jiancheng, Nanjing, China). The activities of the TG (mmol/(g prot))/NEFA (μmol/(g prot))/CAT (U/(mg prot))/SOD (U/(mg prot))/GSH-Px (μmol/(g prot))/MDA (nmol/(g prot)) levels of the supernatants were measured according to instructions from a commercial assay kit (Jiancheng, Nanjing, China) [23].

### 2.12. Statistical Analysis

The experiments were carried out in triplicate and the experimental data were analyzed by SPSS 16.0 (SPSS Inc., Chicago, IL, USA). A one-way analysis of variance (ANOVA) was carried out for statistical analysis, and the data are presented as mean ± standard deviation. When *p* < 0.05, data are considered statistically significant.

## 3. Results and Discussion

To investigate the effect of nanoparticles on *C. elegans*, fundamental physiological indicators, including lifespan, reproduction, and a head swing assay of the nematodes, were assessed. As shown in Figure 1a, the survival curve of the HF group was shifted to the left of the control, indicating that the high-fat diet could reduce the lifespan of the *C. elegans*. However, the treatment of nanoparticles (AXT, ACN NPs and AXT-ACN NPs) could shift the lifespan curve of *C. elegans* to the right (to varying extents). Among these three groups, the nematodes cultured with AXT-ACN NPs had the most extended lifespan of 35 days, which is higher than that of the control group (25 days), indicating their remarkable ability to prolong the lifespan of the nematodes. Further experiments were carried out to investigate whether an increased lifespan was accompanied by improved fertility. As shown in Figure 1b, the total spawn production of *C. elegans* in AXT, ACN NPs and AXT-ACN NPs was increased by 4.27%, 16.37% and 22.06%, respectively, compared to the HF group, demonstrating that treatment with nanoparticles can increase the reproduction of *C. elegans*. Additionally, the entire progeny of AXT-ACN NPs group is close to that of the control group, verifying that AXT-ACN NPs show good ability to improve the fecundity of nematodes. Further, the head-swing numbers of the *C. elegans* were recorded to evaluate the motility of nematodes. As displayed in Figure 1c, the *C. elegans* in the HF group had the least head swings at all time points examined, indicating that the high-fat diet could inhibit the mobility of the nematodes. Fortunately, the mobility of nematodes could be recovered by feeding with nanoparticles, and the nematodes fed with AXT-ACN NPs exhibited the highest number of head swings, proving the AXT-ACN NPs have the best effect on improving physiological activity in nematodes. All these observations indicate that AXT-ACN NPs can promote the longevity, reproductive capacity, and mobility of *C. elegans*.

In order to investigate how the nanoparticles impact the ability of *C. elegans* to withstand heat and oxidative stress, the reactions of the nematodes in each group were evaluated after being subjected to H_2_O_2_ stimulation and heat shock. In the heat stress assays, *C. elegans* in the HF group showed a significantly lower survival rate than that of control group (Figure 2a), showing that high-fat diet is unfavorable to the heat resistance of the nematodes. We found that nanoparticle treatment could significantly shift the lifespan curve of *C. elegans* to the right under heat stress, and we also found that nematodes treated with ACN NPs and AXT-ACN NPs showed an increased lifespan (18 h) compared to that of the HF group (16 h). This result indicates that the nanoparticles can improve the acute oxidation resistance of the nematodes when exposed to high temperatures, and they can also prolong their lifespan. A similar result was noticed in the effect of feeding nanoparticles on the antioxidant stress ability of nematodes. As displayed in Figure 2b, the survival rate of *C. elegans* in the AXT-ACN NPs group under oxidative stress conditions at the 10th hour remained at 46.13%, which is significantly higher than that of HF group (18.84%). Additionally, the maximum lifespan of the nematodes in the HF group was 14 h; however, the maximum lifespan of nematodes treated with AXT-ACN NPs reached 18 h, which is the same as that of the control group, demonstrating that the excellent antioxidant activity of AXT-ACN NPs can enhance antioxidant defense in nematodes [20].

Body fat is primarily stored in intestinal and hypodermal cells in *C. elegans*, which can be visualized clearly by Oil Red O staining [24]. As exhibited in Figure 3b, the nematodes in HF group displayed a significant increase in staining level compared to the worms in the control group (Figure 3a). When the nematodes were fed with AXT (Figure 3c) and ACN NPs (Figure 3d), the staining level was pronouncedly decreased and was close to the staining level of the control group. Remarkably, the nematodes treated with AXT-ACNs NPs (Figure 3e) exhibited a reduction in staining level compared to the control group. These results evince that the prepared nanoparticles can decrease the fat storage of *C. elegans*, and the AXT-ACNs NPs have the best effect on reducing fat accumulation in nematodes.

Lipofuscin, a native blue autofluorescent pigment, has been proposed as a stress and aging biomarker in *C. elegans*, as stress promotes the formation of this pigment due to oxidative degradation and autophagy of cellular components [22,25]. Excessive accumulation of lipofuscin in nematodes may accelerate the aging process by causing oxidative stress [26]. Figure 4a–e display fluorescence microscopy images of nematodes in the control, HF, AXT, ACN NPs and AXT-ACNs NPs groups, respectively. Noticeably, nematodes treated with AXT, ACN NPs and AXT-ACNs NPs exhibited a notable reduction in lipofuscin levels in comparison to the HF group, which suggests that the AXT-ACNs NPs can effectively attenuate lipofuscin accumulation in *C. elegans*.

ROS is the main oxidative stress product when the body is exposed to inferior conditions (e.g., injury, inflammation, high glucose), and over-accumulation of ROS may cause severe deleterious effects on the cells, organs and body, also known as oxidative stress [27,28]. Oxidative stress can induce oxidative damage in biomacromolecules (e.g., mitochondrial DNA, proteins…), resulting in mitochondrial dysfunction, and it is involved in the occurrence and development of multiple diseases such as obesity and hyperlipidemia [29]. In this paper, the generation of ROS in *C. elegans* caused by a high-fat diet was evaluated by using the fluorescent probe DCFH-DA. As shown in Figure 5a,b,f, the relative ROS production of the nematodes in the HF group was enhanced significantly compared with the control group. By contrast, the relative fluorescence intensity of AXT, ACN NPs and AXT-ACN-NPs was decreased by 1.85%, 3.27% and 5.11%, respectively, compared to that of HF group. It has been reported that fat accumulation is associated with higher ROS levels, and fat deposition can be improved by decreasing the accumulation of ROS [30,31]. Consequently, it can be concluded that the AXT-ACN-NPs can scavenge free radicals and alleviate oxidative damage in high-fat nematodes, thereby reducing lipid accumulation in HF nematodes.

To investigate the effects of AXT-ACN NPs on fat accumulation in *C. elegans* further, the content of NEFA in nematodes was measured. NEFA are the product of neutral fat metabolism, and excessive NEFA can produce free radicals [32,33]. The free radicals can react with the lipid bilayer of the cell membrane and cause lipid peroxidation [34]. Plenty of evidence has shown that excessive NEFA can lead to metabolic disorders such as obesity and hyperlipidemia [35,36]. As we know, fatty acids are stored as TG within organisms [37,38]. Accordingly, the TG level can be perceived as an endpoint indicating fat accumulation [39]. Figure 6a displays the TG contents of the nematodes in each group. The content of TG in *C. elegans* was found to be 1.6 times higher in the HF group (3.24 μmol/(g prot)) compared to the control group (1.98 μmol/(g prot)), demonstrating that a high-fat diet can promote the fat accumulation of TG in nematodes. By contrast, the TG content of the AXT, ACN NPs and AXT-ACN NPs groups declined by 0.24, 0.54 and 1.04 μmol/(g prot), respectively. These results validate that the AXT-ACN NPs have best effects on decreasing the high TG level induced by the high-fat diet. Furthermore, the levels of NEFA of *C. elegans* in each group were evaluated. As exhibited in Figure 6b, the level of NEFA remarkably increased with the high-fat diet and can be effectively reduced by feeding with nanoparticles, which coincides with the TG results. To be specific, the NEFA contents of nematodes in the AXT, ACN NPs and AXT-ACN NPs groups decreased by 0.23, 0.38 and 0.58 μmol/(g prot), respectively, compared to that of the HF group. Therefore, it can be inferred that AXT-ACN NPs exhibit a significant capacity to mitigate fat accumulation and decrease the levels of NEFA in high-fat nematodes. This effect can be attributed to the synergistic interplay between anthocyanin and astaxanthin in lipid-lowering mechanisms. Previous studies have demonstrated that anthocyanin extracts effectively reduce fat deposition by down-regulating *sbp-1*, *cebp*, and *hosl-1* (an ortholog of hormone-sensitive lipase homolog) expression while concurrently up-regulating *nhr-49* expression in *C. elegans* [40]. Additionally, astaxanthin has been reported to possess remarkable lipid-lowering properties through modulation of the oleic acid (C18:1Δ9)–stearic acid (C18:0) ratio, thereby reducing overall fat deposition, TG levels, and large lipid droplet formation in nematodes [5]. Furthermore, astaxanthin also exerts control over *sbp-1* and *daf-16* expression while down-regulating *fat-6* and *fat-7* genes. These actions lead to alterations in fatty acid composition within nematodes, as well as inhibition of their fatty acid synthesis process [5].

Antioxidant enzymes play critical roles in ROS scavenging and maintaining cellular redox homeostasis [41]. For instance, SOD can initiate the detoxification of ROS by scavenging O_2_^−^ and converting it to H_2_O_2_, after which CAT and GSH-Px can convert the H_2_O_2_ to H_2_O and a complete antioxidant chain is formed to protect the body from poisoning [42,43,44]. Figure 7a–c present the CAT, SOD and GSH-Px level of the nematodes in each group. It can be clearly observed that the activities of CAT, SOD and GSH-Px of the nematodes declined after being treated with high-fat medium. By feeding with nanoparticles, the activities of these antioxidant enzymes were improved. Notably, the activities of CAT, SOD and GSH-Px of the nematodes fed with AXT-ACN NPs were increased by 14.46 U/(mg prot), 28.58 U/(mg prot) and 10.55 μmol/(g prot), respectively. These results show that the prepared AXT-ACN NPs can improve the activities of the antioxidant enzymes in *C. elegans*. MDA is a pivotal biomarker that can reflect the organ lipid peroxidation rate and intensity of peroxidation damage [45]. As depicted in Figure 7d, the MDA level in the nematodes of the HF group doubled compared to the control group. Additionally, the MDA content of the nematodes declined by 1.76, 2.62 and 3.56 nmol/(g prot) after feeding with AXT, ACN NPs and AXT-ACN NPs, respectively, verifying that AXT-ACN NPs can greatly reduce the MDA content in high-fat nematodes. To sum up, the prepared AXT-ACN NPs can enhance the activity of SOD, CAT and GSH-Px and decrease the MDA level, revealing their great potential in antioxidant and lipid lowering.

## 4. Conclusions

In conclusion, high-fat *C. elegans* induced by a high-glucose diet were used as a model organism to investigate the lipid-lowering and antioxidant effects of AXT-ACN NPs. The results revealed that the maximal lifespan increased by 52.2% to 35 days in *C. elegans* treated with AXT-ACN NPs. By analyzing the head-swinging behavior and total spawn production of the nematodes, we found that AXT-ACN NPs can increase the locomotion behavior and reproductive capacity of high-fat *C. elegans*. Meanwhile, the AXT-ACN NPs also proved be able to reduce the lipid accumulation and the TG, NEFA and MDA contents of high-fat *C. elegans*, verifying their good ability in lipid lowering. Moreover, the AXT-ACN NPs were proven to be effective in decreasing the ROS level and increasing the activities of CAT, SOD and GSH-Px, confirming their excellent antioxidant ability. The findings of this study imply that AXT-ACN NPs hold significant potential for application in functional foods. To establish a connection between research and practical use, further investigations involving organoid strategy and in vivo experiments will be utilized to assess the lipid-lowering and antioxidant effects of AXT-ACN NPs.

## Figures and Tables

**Figure 1 foods-13-00514-f001:**
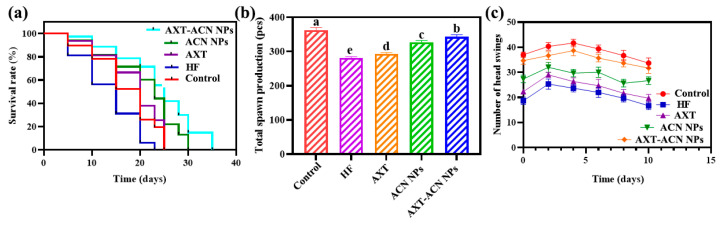
Effects of astaxanthin nanoparticles on (**a**) life span, (**b**) total oviposition, and (**c**) head swing times of high-fat *C. elegans*. Different letters indicate significant difference (*p* < 0.05).

**Figure 2 foods-13-00514-f002:**
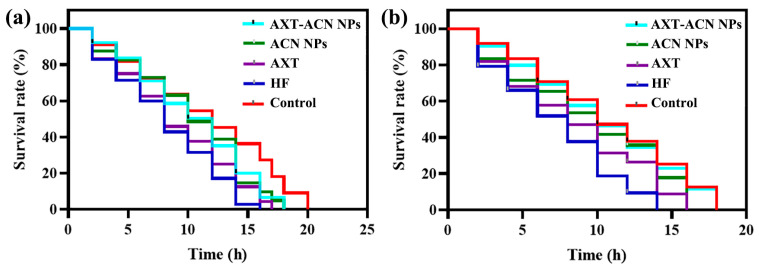
Effects of astaxanthin nanoparticles on the survival rate of high-fat *C. elegans* under (**a**) heat stress and (**b**) oxidative stress.

**Figure 3 foods-13-00514-f003:**
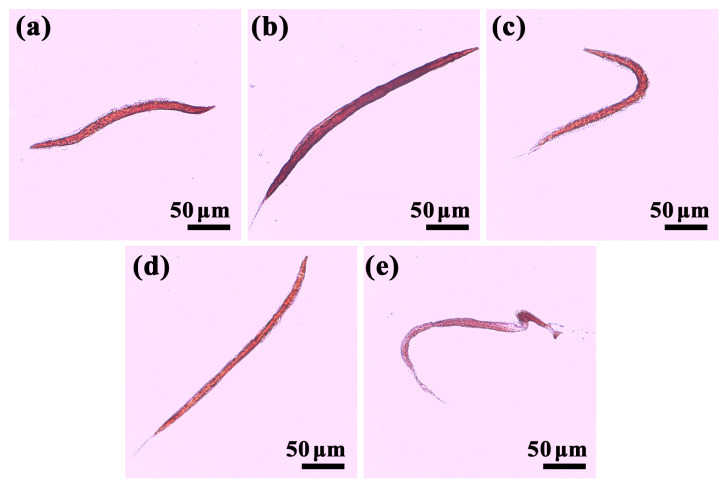
The effect of astaxanthin nanoparticles on the fat content of high-fat *C. elegans* was observed by Oil Red O staining ((**a**): control group; (**b**): HF group; (**c**): AXT group; (**d**): ACN NPs group; (**e**): AXT-ACN NPs group).

**Figure 4 foods-13-00514-f004:**
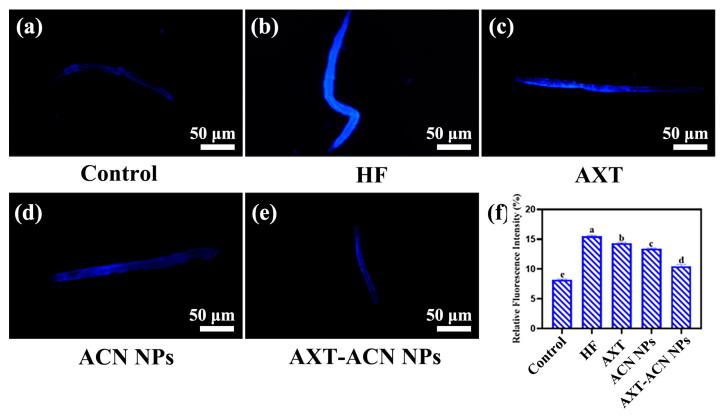
Effect of astaxanthin nanoparticles on lipofuscin content in high-fat *C. elegans* ((**a**): control group; (**b**): HF group; (**c**): AXT group; (**d**): ACN NPs group; (**e**): AXT-ACN NPs group; (**f**): Average fluorescence intensity of each group). Different letters indicate significant difference (*p* < 0.05).

**Figure 5 foods-13-00514-f005:**
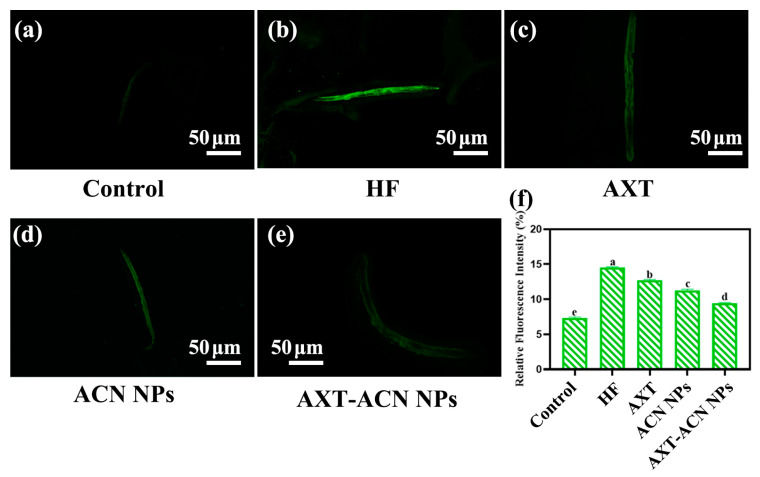
Effects of astaxanthin nanoparticles on ROS levels in high-fat *C. elegans* ((**a**): control group; (**b**): HF group; (**c**): AXT group; (**d**): ACN NPs group; (**e**): AXT-ACN NPs group; (**f**): average fluorescence intensity of each group). Different letters indicate significant difference (*p* < 0.05).

**Figure 6 foods-13-00514-f006:**
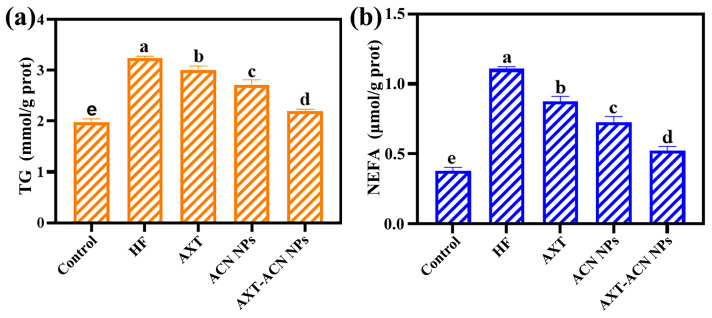
Effects of the astaxanthin nano-delivery system on contents of (**a**) TG and (**b**) NEFA in high-fat nematodes. Different letters indicate significant difference (*p* < 0.05).

**Figure 7 foods-13-00514-f007:**
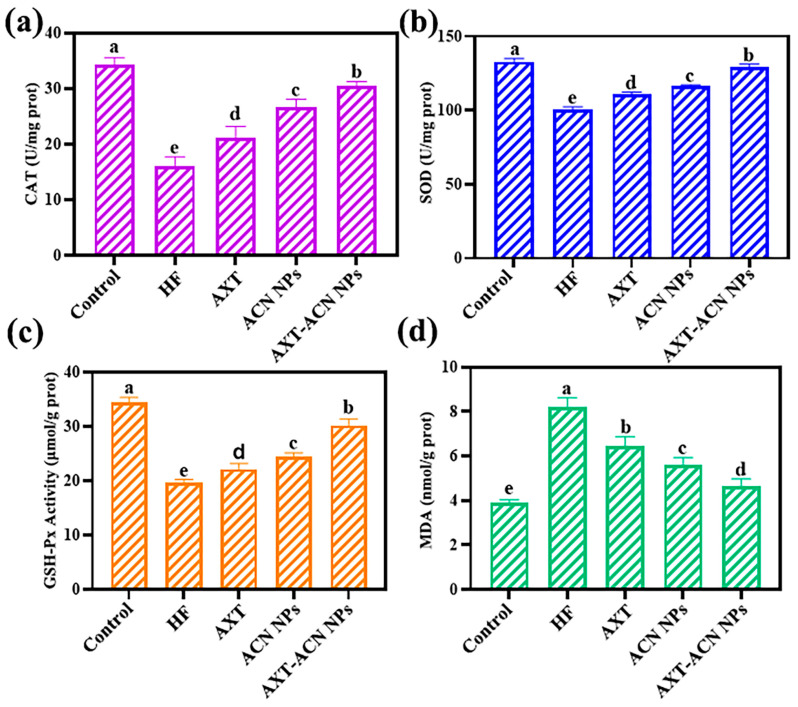
Effects of astaxanthin nano-delivery System on the contents of (**a**) CAT, (**b**) SOD, (**c**) GSH-Px and (**d**) MDA in high-fat nematodes. Different letters indicate significant difference (*p* < 0.05).

## Data Availability

Data is contained within the article, further inquiries can be directed to the corresponding author.

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
