# Peer review of "Lipid-Lowering and Antioxidant Effects of Self-Assembled Astaxanthin–Anthocyanin Nanoparticles on High-Fat Caenorhabditis elegans"

_foods, 2024, doi:10.3390/foods13040514_

Round 1

Reviewer 1 Report

Comments and Suggestions for Authors

The study was presented well in terms of how the various components were presented, beginning with an introduction that provided adequate context and included all pertinent references. The study's findings were compared with previously published findings and presented and debated in a solid scientific way. The presentation concluded with scholarly references relevant to the research topic. As a result, the research design is arranged correctly.

1- page 1line 16-28: The abstract need to improve by adding more interesting results 

2-  page 2 line 66-74: the aim of the work [Herein, high-fat C. elegans were cultured by feeding a high-sugar diet and further used as a model organism to evaluate the impact of AXT-ACN NPs on the lifespan, head swinging behavior and total spawn production. Subsequently, the effect of astaxanthin nanoparticles on fat content, lipofuscin content and ROS level of high-fat nematodes was investigated via microscopic methods. In the end, the content of triglyceride (TG), non- esterified free fatty acids (NEFA), and antioxidant-related enzymes in nematodes were carefully studied, which validated that the AXT-ACN NPs can reduce the content of free fat acids in high-fat nematodes, promote fat hydrolysis, inhibit fatty acid synthesis, reduce fat accumulation and show great antioxidant ability, making them as facile food-grade supplements as alternatives to weight-loss medications.] need to rewrite to be clear understand  for any one.

3- page 3 line 120: the word and should be omit.

4- page 3,4 lines starting from 120 ----etc :  all methods starting from 

  2.5 Reproduction assay ,

 2.6 Head swing assay , 

2.7 Stress resistance assay, 

2.8 Oil Red O staining of C. elegans,

 2.9 Measurement of lipofuscin accumulation,

 2.10 Determination of the reactive oxygen species (ROS) level, 

2.11 Measurement of TG, NEFA, GSH-PX, SOD, CAT and MDA level,  

  reference should be added  for each method.

Comments on the Quality of English Language

 Minor editing of English language required

Author Response

We have revised the manuscript according to your nice advice.

Reviewer 2 Report

Comments and Suggestions for Authors

The ms is very well written in that the authors focus on the lipid-lowering and antioxidant effects of the astaxanthin-anthocyanin nanoparticles on high-fat Caenorhabditis elegans. All observations are experimentally documented. All appropriate controls are used and conclusions are formulated according to the nature of the results.  Especially the observed results that the AXT-ACN NPs can enhance the antioxidant defense in nematodes are of great importance and raise many intriguing questions about the possible cellular mechanisms. It might sound speculative but it would be interesting to discuss in a short paragraph the lipid-lowing effect and to indicate in what direction the authors would continue their research regarding the AXT-ACN NPs.

Author Response

We have revised the manuscript according to your nice advice. Please see the attachment.

Reviewer 3 Report

Comments and Suggestions for Authors

Manuscript titled “Lipid-lowering and antioxidant effects of self-assembled astaxanthin-anthocyanin nanoparticles on high-fat Caenorhabditis elegans” reports the effects of astaxanthin and anthocyanin nanoparticles on C. elegans, as a potential treatment to counter lipid accumulation and to validate their effects on related physiological markers. The experiment is based on previous work by the authors, although there are some issues that require improvement. One of the most significant criticisms is that the discussion mostly mentions the results obtained herein and differences between groups, but minimal or no comparisons with other works are provided. Moreover, potential mechanisms of action to explain the observed results are also not considered. Discussing your findings as compared to those of other authors and providing potential explanations for the observed effects, rather than just stating your findings, can provide vast support for your work. Both astaxanthin and anthocyanin have been widely studied, thus, there is ample evidence which could provide this support.

Also see specific comments:

1.       In line 49, “conjugated unsaturated double bond” appears to be redundant, perhaps “conjugated double bonds” would be more appropriate.

2.       In section 2.3, the treatments and times are not immediately clear. For example, only 5 days are mentioned for glucose, but no times and doses are clearly mentioned for the nanoparticle-treated groups.

3.       The manuscript mentions “high fat”, including the title, although section 2.3 states that “10 mM glucose” (line 111) was used instead. Shouldn’t the work be based on a high-glucose or high-carbohydrate model? Although carbohydrates lead to fat accumulation, the treatment is based on glucose and not on fats. Can the authors please comment on this?

4.       The title of section 2.5 appears to be missing something (“Reproduction assay and”).

5.       In section 2.11, please define the enzyme units for GSH-PX, SOD and CAT. Moreover, these values are expressed per unit of protein, although the assay used to quantify protein is not stated; please add it.

6.       The abstract (line 17) and results and discussion (196, 204 and 280) mention “as-prepared” nanoparticles. What do you mean exactly by “as-prepared”? It is not entirely clear why this phrase needs to be mentioned.

7.       A comparison of TG levels is mentioned in line 276 as “1.5 times higher”, while line 279 then states declines of “0.24, 0.54 and 1.04 μmol/gprot”. Please consider discussing these differences using the same units, for example, all as X times higher or lower than the control, or X umol higher or lower than the control. This will make the differences between groups easier to understand.

8.       Line 288 states that the treatments “promote fat hydrolysis, inhibit fatty acid synthesis, and reduce fat accumulation”. The data of the manuscript only supports a reduced fat accumulation, but promoting fat hydrolysis or inhibiting fatty acid synthesis cannot be directly confirmed nor denied. Although both may be true, it cannot be definitely stated. Please consider rephrasing.

9.       In lines 279 and 286 and figure 6, please consider adding a space between “g” and “prot” or changing to “μmol/g of protein” for increased clarity. Likewise for figure 7 “gprot” and “mgprot”. In figure 7(c) and line 305, please confirm if it should be “mmol” or “μmol” (or something else).

Comments on the Quality of English Language

10.       Some minor issues could be fixed, but the information is understandable.

Author Response

(The authors gave the same response as above.)

Reviewer 4 Report

Comments and Suggestions for Authors

First of all, there is a real effort put into doing this research. I also felt that I enjoyed and wanted to continue reading this research. but i have afew comments:

1. The abstract is full of abbreviations without any definition, it willbe good if you define the abbriviation for the first appearance

2. Also, it will be better if it replaces the keywords with other than those found in the title

3. You mentioned that made the HF groupe by feeding on 10 mM of glucose for 5 days, is that considered a high fat diet, please explane that 

4. At title 2.3: you didn't show the dose of AXT, ACN... etc that you used for treating of C. alegance, you mentioned only that you mixed them with E.coli . Also, you didn't show if you treat with them daily or what and what the importance of E.coli?

5.At title 2.5: Production assay and ... what? complete the title, please

6. You need to show the chemical composition of the NGM plate 

7. It will be better if you cited all references you used in the methods part 

8. Results and discussions are written well but need more explanation for each effect of Axt and ACN 

Comments on the Quality of English Language

Minor editing of English language required

Author Response

(The authors gave the same response as above.)

Round 2

Reviewer 3 Report

Comments and Suggestions for Authors

Manuscript titled “Lipid-lowering and antioxidant effects of self-assembled astaxanthin-anthocyanin nanoparticles on high-fat Caenorhabditis elegans” reports the effects of astaxanthin and anthocyanin nanoparticles on C. elegans, as a potential treatment to counter lipid accumulation and to validate their effects on related physiological markers.

The present version of the manuscript was modified according to comments and suggestions made during an initial revision. The most significant issue raised by the present reviewer includes a lack of discussion/comparisons between the authors’ work and that of others, since the manuscript mainly describes the results reported. This issue remains mostly unaddressed, since only a few lines were added to the discussion (309-321), while most findings remain without a proper substantial discussion. Although the authors’ work is notable, it remains mostly descriptive, which is its main drawback. Additional comments made were:

1. Rephrasing a potentially redundant phrase in the introduction. The phrase was modified.

2. In section 2.3, providing a clearer description of the times and treatments used. Doses and treatments are now mentioned.

3. Commenting on referring to the model as “high fat”, when the treatment is based on glucose. The authors mention that high carbohydrate diets lead to the development of obesity and other conditions. I completely agree with this observation, the use of the model and its validity, although the issue is with how the model/experiment is named, since referring to the work as “high fat” is not accurate, since it is based on administering a high glucose dose to the animals. Although I disagree with the authors on this, I will not insist on rephrasing.

4. Adding some apparently missing information to the title of section 2.5. The authors mention that “and” was deleted from the heading; however, it was apparently missed. This is not a significant issue, and can be fixed during final editing.

5. Defining the enzyme units for GSH-PX, SOD and CAT, and stating the method used to quantify protein. BCA has now been mentioned as the method used to quantify protein, however, the units of the enzymes have still not been provided, the authors only state that they followed manufacturer’s instructions. Unit information can be useful for other authors who may want to compare their work with the present one, since a unit can be defined in various ways, and clearly specifying them can remove unnecessary ambiguity.

6. Clarifying the use of the term “as-prepared”. This has been removed.

7. Homogenizing a comparison between changes in values (X times higher) and absolute units (X umol/g prot). The comparison has been homogenized.

8. Rephrasing to only state that the treatments reduced fat accumulation, without including unsupported arguments. The phrase was adequately modified.

9. Assing a space in the units between “g” and “prot”. Units have been adequately modified.

According to the aforementioned changes made by the authors, it is apparent that most issues raised by the present reviewer were addressed, although some still remain. The only significant issues that remain are a lack of comparisons between the authors’ work and those of others and/or more in-depth explanations for their findings, in addition to properly defining the enzymes’ units (comment 5). Please consider making these improvements to your work if possible.

Comments on the Quality of English Language

Some minor issues, but the information is understandable.

Author Response

Thanks for your nice suggestion. We have included the unit of enzymes in our revised manuscript and other typo errors. You can find the response letter in the attachment.
